# A Three Demultiplexer C-Band Using Angled Multimode Interference in GaN–SiO_2_ Slot Waveguide Structures

**DOI:** 10.3390/nano10122338

**Published:** 2020-11-25

**Authors:** Eduard Ioudashkin, Dror Malka

**Affiliations:** Faculty of Engineering, Holon Institute of Technology (HIT), Holon 5810201, Israel; strooperfb@gmail.com

**Keywords:** demultiplexer, AMMI, slot-waveguide, GaN, C-band

## Abstract

One of the most common techniques for increasing data bitrate using the telecommunication system is to use dense wavelength division multiplexing (DWDM). However, the implementation of DWDM with more channels requires additional waveguide coupler devices and greater energy consumption, which can limit the system performances. To solve these issues, we propose a new approach for designing the demultiplexer using angled multimode interference (AMMI) in gallium nitride (GaN)–silica (SiO_2_) slot waveguide structures. SiO_2_ and GaN materials are selected for confining the infrared light inside the GaN areas under the transverse electric (TE) field mode. The results show that, after 3.56 mm light propagation, three infrared wavelengths in the C-band can be demultiplexed using a single AMMI coupler with a power loss of 1.31 to 2.44 dB, large bandwidth of 12 to 13.69 nm, very low power back reflection of 47.64 to 48.76 dB, and crosstalk of −12.67 to −15.62 dB. Thus, the proposed design has the potential for improving performances in the telecommunication system that works with DWDM technology.

## 1. Introduction

Optical demultiplexing devices are key elements in the optical telecommunication system that works with dense wavelength division multiplexing (DWDM) and serve the purpose of increasing the data transfer capacity for the enlarged number of end-users [1]. DWDM apparatus being investigated constantly includes arrayed waveguide gratings (AWGs) [2], multimode interference (MMI) devices [3], photonic crystal fiber [4], and angled MMI (AMMI) [5].

Multimode interference device operation is based on the self-imaging principle, in which the outline of the input field is replicated along the propagation axis of the waveguide at periodical points [6]. MMI devices are incorporated into photonic integrated circuits (PIC) thanks to their low excess loss, large optical bandwidth, and plain structure [7,8]. AMMI is a form of MMI device in which the single-mode access waveguides are tilted in the desired angle with respect to the propagation axis of the multi-mode MMI core. The tilted inputs/outputs allow the design of a single AMMI demultiplexer with more channels (>2) compared with one MMI coupler. Power back reflection is propagating light that reflects back into the input source owing to the refractive index mismatch at the boundaries between different materials and a side effect of the self-imaging effect [9]. In terms of losses, MMI or AWG are inferior to the AMMI in the case of back reflection that is inherent in their architecture [10]. Slot waveguide structure as introduced by Almeida et al. [11] is a novel waveguide mechanism that is applied for splitters [12], AMMIs [13], and other applications and can be utilized using an already existing and sufficient manufacturing process in the silicon on insulator (SOI) method [14]. In a slot waveguide, a layer of a low refractive index material (slot area) is surrounded by two layers of high refractive index materials and the guidance of light is the result of the total internal reflection (TIR) effect [15]. In addition, because of the strong power confinement inside the slot area, there are no confinement losses in the slot waveguide structure [16].

Gallium nitride (GaN) is a transparent semiconductor material with notable features such as superior electrical properties, resistance to temperature [17], and a wide spectral range. Recent studies show the gain of using GaN-based conventional waveguides for optical splitters [18] or couplers [19]. Silica (SiO_2_) is a clear glass-type low-absorption insulator that is not moisture-absorbing nor permeable to gasses [20] and has low reactivity with acids. MMI devices based on slot waveguides technology are sensitive to the contrast in the refractive indices, which can affect their performance [21]. Hence, it is better to use GaN–SiO_2_ slot waveguide, which has a smaller difference in the refractive index compared with other materials combinations (e.g., silicon–SiO_2_ [22]).

The horizontal slot waveguide manufacturing process applied by lithography patterning and a multilayered deposition or thermal oxidation is superior to the vertical slot waveguide in terms of lower interface roughness, which can lead to a decrease of scattering loss. The horizontal structure also has little to no constraints on the waveguide thickness with layers up to a nanometer precision [23,24]. SOI nanophotonic devices may be fabricated on a BOX layer on top of a Si wafer substrate [25] with GaN–SiO_2_ stacking such as demultiplexers [15] or splitters [21]. The multilayering process of the waveguide and the protective cladding consists of an e-beam lithography patterning, thin film deposition of GaN and SiO_2_ using plasma enhanced chemical vapor deposition, alongside cleaning and etching of layer surfaces [26,27,28].

In this paper, we introduce an innovative design of a three-channel AMMI based on the GaN–SiO_2_ slot waveguide structure. The device functions as a 1 × 3 demultiplexer with operating wavelengths of 1.53, 1.5475, and 1.565 µm. Numerical investigations were carried out on the geometrical criteria of the device to find the optimal angle for the input/outputs waveguide taper. Moreover, the self-imaging effect in the MMI coupler and strong power confinement inside the GaN slot waveguide areas are achieved for a highly desired power at the outputs.

The simulations and the optimizations were carried out using the full vector beam propagation method (FV-BPM) [29] alongside MATLAB script codes.

The novelty of using GaN–SiO_2_ slot waveguide technology is the ability to manipulate the light propagation in the MMI section for obtaining a lower beat length for small channel spacing and low back reflection (BR) at the input source.

The proposed device in this work can be used in a DWDM system for increasing the communication data bitrate.

## 2. Slot Waveguide AMMI Structure and Theoretical Aspect

The horizontal slot waveguide structure consists of a slot region that rests between two slab layers and its schematic sketch at the x–y plane is shown in Figure 1. The slot area is represented in gray color, which is SiO_2_ material with refractive index n_S_ and thickness layer of H_S_. The slab layers are shown in green color for GaN material with refractive index n_H_ and thickness layer of H_H_. The surrounding cladding area is shown in gray color, which is SiO_2_ material. The slot waveguide width is W and the total thickness of the slot waveguide is H_T_.

Figure 2 shows the 1 × 3 AMMI demultiplexer structure at the x–z plane. The 1 × 3 demultiplexer is based on one MMI coupler with 1/3 tilted angle input/output waveguide tapers accordingly. The MMI coupler width is W_MMI_ and the length is L_MMI_. Each input or output taper waveguide is titled with an angle θ_t_ and is tapered from W_t,1_ (single mode width) to W_t,2_ (multimode width) over a length of L_t_. 

The optimal geometrical parameters of the AMMI demultiplexer are W_MMI_ = 20.5 µm, L_MMI_ = 3.56 mm, length of the taper of 120 µm with width variation between 0.5 and 42.4 µm, and an optimal tilt angle of the tapers of 0.3 rad. The output channels are tagged as Ch_i_ and their lengths along the propagation direction (z-axis) are given by L_i_ (i = A, B, C), as summarized in Table 1. The gap between channel A and channel B was set to 127 μm and the gap between channel B and channel C was set to 132 μm.

In a slot waveguide, the discontinuity of the electric field in the boundaries between the high and low refractive index materials is the reason for obtaining strong light confinement. The amplitudes ratio (AR) of the electric field inside/outside the boundary is a function of the refractive indices of the materials inside/outside the boundaries, respectively. This relation is described in Equation (1), where a represents the value of the axis for the boundary [11].
(1)$$AR=\frac{E_{\left|x\right|=a-}}{E_{\left|x\right|=a+}}=\frac{n_{H}^{2}}{n_{S}^{2}},$$

In our design, the AR and the refractive indices (n_S_ and n_H_) for the chosen slot waveguide materials as a function of the operating wavelengths are shown in Table 2.

The access waveguides are placed along the propagation region to satisfy the periodic self-imaging and TIR effects. These mechanisms allow guiding the input signal to the desired output channel, which is located at different points along the z-axis (L_i_) and can be estimated by Equation (2) [5].
(2)$$L_{i}\approx \frac{4n_{\mathrm{eff}}w_{\mathrm{eff}}^{2}}{\lambda_{i}},\begin{array}{cc} & i=A,B,C \end{array}$$
where n_eff_ is the effective refractive index of the major mode; λ_i_ is the operating wavelength; and W_eff_ is the effective width of the MMI coupler, which incorporates the lmbert Fedorov effect, as stated in Equation (3) in the case of transverse electric (TE) mode [30].
(3)$$W_{\mathrm{eff}}=W_{\mathrm{MMI}}+\frac{\lambda_{i}}{\pi }{\left(n_{\mathrm{eff}}^{2}-n_{S}^{2}\right)}^{-1/2}$$

The crosstalk is given by Equation (4).
(4)$$\mathrm{Crosstalk}(\mathrm{dB})=10\log\left(\frac{p_{m}}{p_{n}}\right)$$
where P_n_ is the desired power from the transmission channel and P_m_ is the interference power from the neighboring channels.

The insertion loss (IL) at the channels output contributes to the examination of the device performance and is given by Equation (5).
(5)$$\mathrm{IL}(\mathrm{dB})=-10\log\left(\frac{P_{\mathrm{out}}}{P_{\mathrm{in}}}\right)$$
where P_in_ is the input taper power and P_out_ is the power at the output taper channel. 

## 3. Simulation Results 

The slot waveguide structure was solved using FV-BPM simulations combined with MATLAB script codes, which were used to analyze and set the optimal geometrical parameters. From these simulations, Figure 3a,b and Figure 4a,b, the optimal slot thickness parameters are H_H_ = 300 nm, H_S_ = 75 nm, H_T_ = 675 nm, and W = W_t,1_ = 500 nm. These geometric values were chosen to support strong light confinement in the GaN–SiO_2_ slot waveguide during light propagation and are the result of optimizations performed using the FV-BPM solver.

Figure 3a,b shows the normalized power as a function of the GaN (H_H_) and SiO_2_ (H_S_) layer thickness at 1.53 μm wavelength. From Figure 3a,b, the tolerance ranges for the GaN and SiO_2_ thickness layer that ensure strong power confinement (above 90% of the normalized power) inside the slot waveguide can be found and their values are 270–328 nm and 10–119 nm, respectively.

Figure 4a shows the fundamental quasi-TE mode solution at the x–y plane for a wavelength of 1.53 µm. It can be observed that the light is highly confined inside the GaN layers (red color). The red color in the slab (GaN) areas indicates a strong intensity near the boundaries between the slot and the slab areas, as expected from theory. Figure 4b shows the vertical cut at x = 0 nm and the normalized power level for the slot waveguide thickness over the y-axis. The wide tolerance range of the high-power level over the y-axis (around 700 nm) can be noticed in Figure 4b. This range can be very helpful for utilizing better adjustments between the transmission signal and the waveguide input taper. Figure 4c shows the existence of a quasi-TM polarized mode solution at the x–y plane of the suggested device for a wavelength of 1.53 µm. This solution shows strong light confinement inside the slot area (SiO_2_) and further investigations and simulations can be carried out on the quasi-TM polarization with relevant geometry adjustments. However, in this study, the proposed device was set to the quasi-TE polarization for obtaining a single-mode transmission.

Figure 5 shows the normalized power as a function of the tilt angle of the input and output tapered waveguides (θ_t_) for the operating wavelengths. The chosen value for θ_t_ was set to achieve maximum power at the output tapers and very low BR. In our design, the optimal θ_t_ value is set to be 0.3 rad. In this figure, it can be seen that channels A and B are less sensitive to the fabrication tilt error of ±0.01 rad (±0.5 degree) with losses of 0.4 and 0.8 dB, respectively, and channel C is more sensitive, with losses of 1.4 dB for the same error.

Figure 6 shows the optimization process for the MMI coupler width (W_MMI_) and it is based on an estimation performed using Equation (3) with FV-BPM simulations. The W_MMI_ optimal value was selected to be 20.5 µm to ensure the highest output power for all channels and the best tolerance ranges for fabrication errors. In this figure, it can be seen that channels A, B, and C are robust to the fabrication error for the MMI coupler width of ±20 nm from the optimal value with losses of 0.47, 0.75, and 0.96 dB, respectively.

Figure 7a–c shows the normalized power for each output taper for the operating wavelengths as a function of the locations along the z-axis of the output channels L_A_, L_B_, and L_c_, accordingly, as stated in Table 2. The location of a channel directly affects the power distribution between the channels. The individual outputs are located to satisfy the beat length requirements of the chosen quasi-TE polarization for a chosen wavelength and polarization deviations caused by spatial shift may cause an abrupt power variation in the output, as seen in Figure 7b. Therefore, it is important to clarify that the selected optimal values were chosen to satisfy the highest output power for all three channels. In this figure, it can be seen that channels A, B, and C have a low sensitivity to fabrication error for the MMI coupler length of ±1 μm from the optimal value with losses of 0.08, 0.05, and 0.1 dB, respectively.

Figure 8a–c shows the light propagation of the operating wavelengths from the input taper to the desired output taper (channel) through the MMI coupler. Figure 8a shows the light transmission of 1.53 μm wavelength from the input tilt taper to the output taper Ch_A_ at 3.56 mm. Figure 8b shows the light transmission of 1.5475 μm wavelength from the input tilt taper to the output taper Ch_B_ at 3.39 mm. Figure 8c shows the light transmission of 1.565 μm wavelength from the input tilt taper to the output taper Ch_C_ at 3.215 mm. From these figures, it can be seen that low losses occur between the output tapered channels (blue color) and inside the MMI coupler (blue color).

Figure 9 shows the optical transmittance of the AMMI device for wavelengths in and around the C-band range (1522–1573 nm) as a result of FV-BPM simulations combined with MATLAB script codes. From Figure 9 combined with Equations (4) and (5), the values of the IL, crosstalk, and full width at half maximum (FWHM) for each output channel are obtained as shown in Table 3.

Figure 10a,b show the normalized power as a function of GaN (H_H_) and SiO_2_ (H_S_) layer thickness for the operating wavelengths. The chosen values were set to achieve maximum power at the output taper. In Figure 10a, it can be seen that channels A, B, and C have a sensitivity to fabrication error for the GaN thickness of ±10 nm from the optimal value with losses of 1.05, 1.17, and 2 dB, respectively. In Figure 10b, it can be seen that channels A, B, and C have a sensitivity to fabrication error for the SiO_2_ thickness of ±5 nm from the optimal value with losses of 1.15, 1.38, and 2.14 dB, respectively.

Another important property of the device is the BR, which can affect the telecommunication system, especially in the case of reflections that are propagated in the reverse light directly into the laser source. The MMI coupler can suffer from reflections because of the effective index mismatch and the self-imaging effect. In our design, reflections can happen at the interfaces between the MMI coupler (GaN–SiO_2_) and the SiO_2_ cladding. To solve this issue, we have integrated angled input and outputs waveguide tapers that are used to minimize reflections. Using a finite difference time domain (FDTD) algorithm, the power level of light reflections coming from the MMI coupler for each operating wavelength can be found. A monitor was located in the input waveguide taper to collect the light reflections coming from the MMI coupler, as shown in Figure 11a. Figure 11b shows the power BR as a function of cT (optical path length) for 1.53 µm wavelength. The optical path length was set to 700 µm to include all reflections coming from the MMI coupler owing to the self-imaging effect. Table 4 shows that reflection losses can be neglected because of the lowest average power BR (48.03 dB) for the operating wavelengths. This is a major advance compared with a standard MMI coupler that usually suffers from BR losses.

To study the advantages of using AMMI-based GaN slot-waveguide technology, several C-band/O-band AMMI demultiplexers that were previously published and currently used were compared to this work, as shown in Table 5. The attributes compared incorporate waveguide technology (WT), number of channels (NOC), channels spacing (CS) in terms of spectral range, footprint (width × length), maximum insertion losses (IL), best crosstalk (XT), maximum BR, and year of publication.

The proposed design has two main benefits compared with the other AMMI demultiplexer devices: small CS and low BR. These advantages are because GaN–SiO_2_ slot-waveguide technology can obtain a small effective index in the MMI section, which leads to a very low BR and low beat length. The BR property can be utilized especially for the communication system that works with a laser source that suffered from BR.

## 4. GaN Fabrication Losses

GaN (stack) waveguide losses from the fabrication process consisting of scattering loss, free carriers loss, and crystallization level impact are discussed later in this paper and are described in detail by Chen, H [32], and Al-Zouhbi [33].

Propagation losses in the GaN layers are influenced directly from the practical fabrication of the layers and should be considered prior to manufacturing of the suggested device. Some of the contributors to the propagation losses are surface roughness due to surface roughness and free carrier loss [32] alongside the crystallization level of GaN [33]. The scattering loss is expected to be higher than that stated in [32] (~2.5 dB/cm) because of the large device dimensions, despite the large wavelength. The intrinsically n-type doped GaN may possess a high-density n-type free carrier (up to 10^17^ cm^−3^), which will contribute to the free carrier loss of more than 1 dB/cm [32]. To lower these losses, surface roughness can be improved by better controlling the dry etching speeds at grain boundaries or by small power applications. Amorphous GaN in comparison with crystalline GaN has the advantages of lower manufacturing costs, better suitability for deposition on various substrates, while relieving the difficulty of lattice matching of the Ga-N bond; in contrast, however, it may additionally have significant material absorption of up to 13 dB/cm [33]. Recently, researchers show the ability to fabricate GaN rib waveguides on Ga_2_O_3_ substrate, using a wet-etch process for improving side-wall verticality and reduction of waveguide surface roughness. The propagation loss in the GaN waveguides has been experimentally determined to be 7.5 dB/cm at the C-band spectrum [34]. Using this fabrication technique, the propagation losses can be estimated to be around 0.75 dB/mm. Thus, the fabrication losses for the proposed device in an actual system for the three channels Ch_A_, Ch_B_, and Ch_C_ are estimated to be 2.67, 2.54, and 2.41 dB, respectively.

## 5. Conclusions

This paper demonstrates a novel and new design of a 1 × 3 AMMI demultiplexer device based on GaN–SiO_2_ slot waveguide structures operating in the C-band spectrum range. The study of the light coupling between the tilt angled input and output waveguide taper to the MMI coupler using the self-imaging effect shows the ability to obtain a higher power level in the output channels by setting the suitable tilt angle. We proved that it is possible to use a single AMMI coupler to divide three channels with a very low power back reflection (47.64 to 48.76 dB) in comparison with a single standard MMI coupler that can divide only two channels with suffering from high power back reflection. These advantages can be utilized for reducing footprint size and total power back reflection for improving commination system performances.

The results show that three operated wavelengths of 1.53, 1.5475, and 1.565 μm can be demultiplexing with low IL of 1.31 to 2.44 dB, good crosstalk of −12.67 to −15.62 dB, and large bandwidth of 12 to 13.69 nm. The quasi-TE fundamental mode solution has a wide range of 700 nm with a high intensity level over the y-axis that can be utilized for reducing coupling losses due to the mismatch modes between the transmission signal to the tilt angle input taper waveguide.

In addition, numerical investigations and optimizations of the key geometrical parameters presented a robust device that has good tolerance ranges for fabrication errors for up to 1.4 dB for the access waveguides tilt angle, up to 0.96 dB for slot waveguide thickness, and up to 0.1 dB for channels’ location, while maintaining a footprint in the range of similar devices of approximately 0.07585 mm^2^ (width × length). In conclusion, the design has great potential for improving performances in the telecommunication system that works with DWDM technology. 

Although only the 1 × 3 wavelength GaN–SiO_2_ AMMI demultiplexer configuration is considered in this paper, the demultiplexer can operate as a 3 × 1 multiplexer with a reversed direction of the guided light. It is also possible to proceed to manufacture considering the purposed device configuration or to test other combinations of materials.

## Figures and Tables

**Figure 1 nanomaterials-10-02338-f001:**
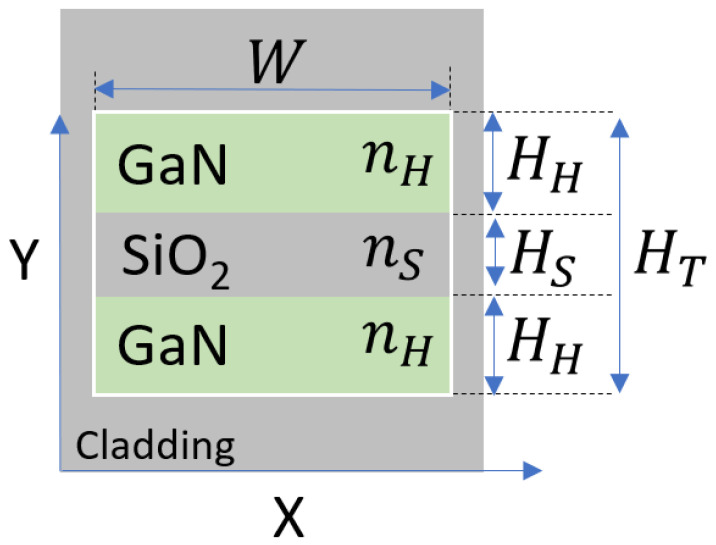
Schematic sketch of slot waveguide structure at the x–y plane.

**Figure 2 nanomaterials-10-02338-f002:**
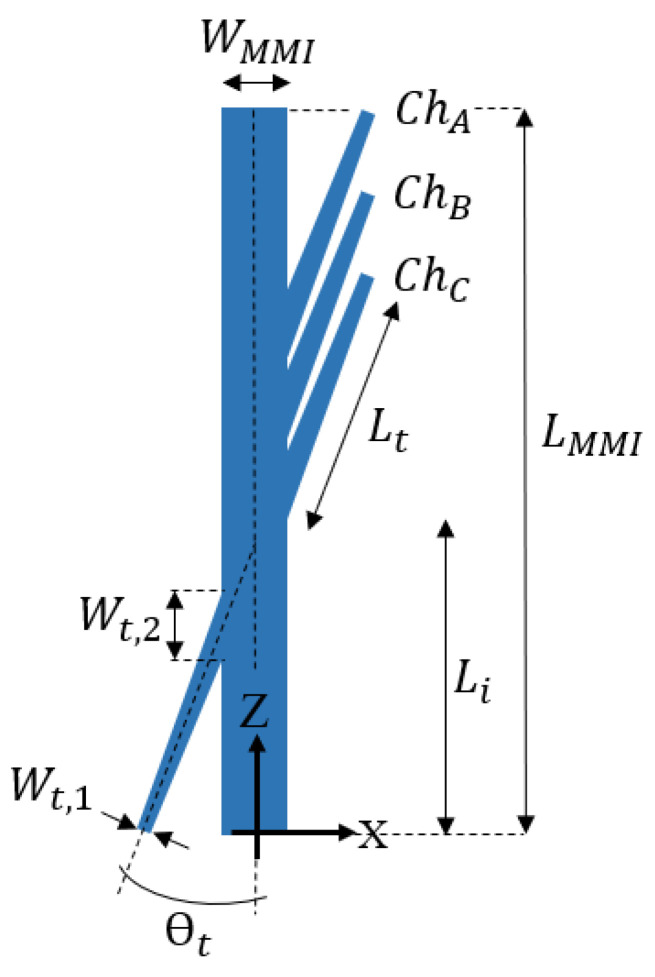
Schematic sketch of the 1 × 3 angled multimode interference (AMMI) demultiplexer at the x–z plane.

**Figure 3 nanomaterials-10-02338-f003:**
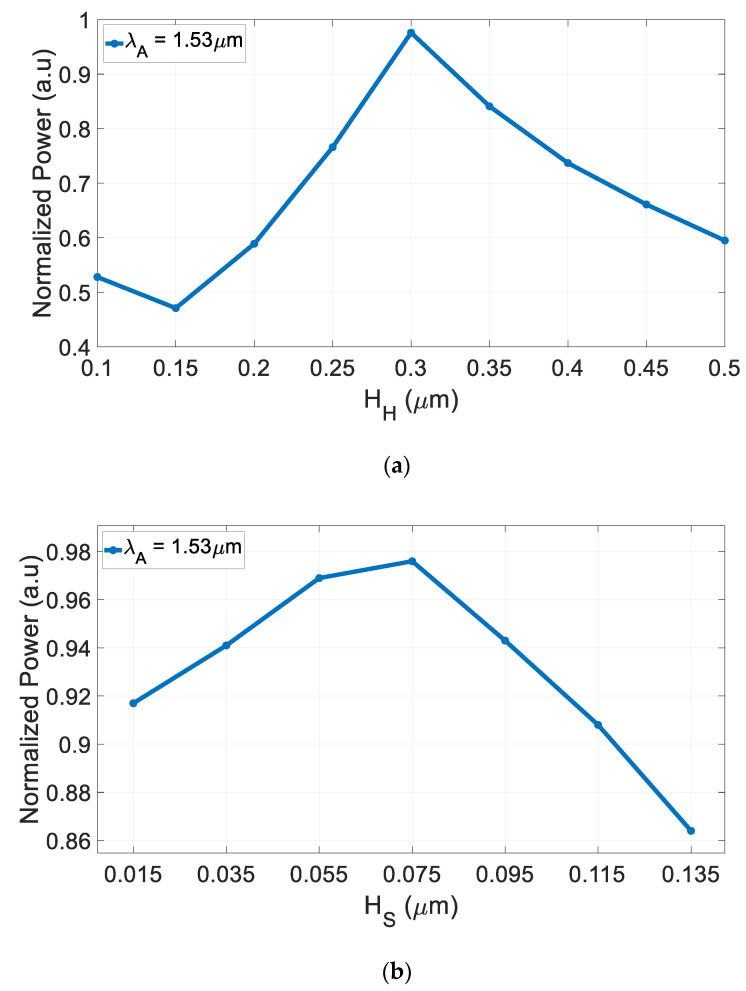
Normalized power as a function of the slot waveguide thickness at 1.53 μm wavelength. (**a**) H_H_ and (**b**) H_S_.

**Figure 4 nanomaterials-10-02338-f004:**
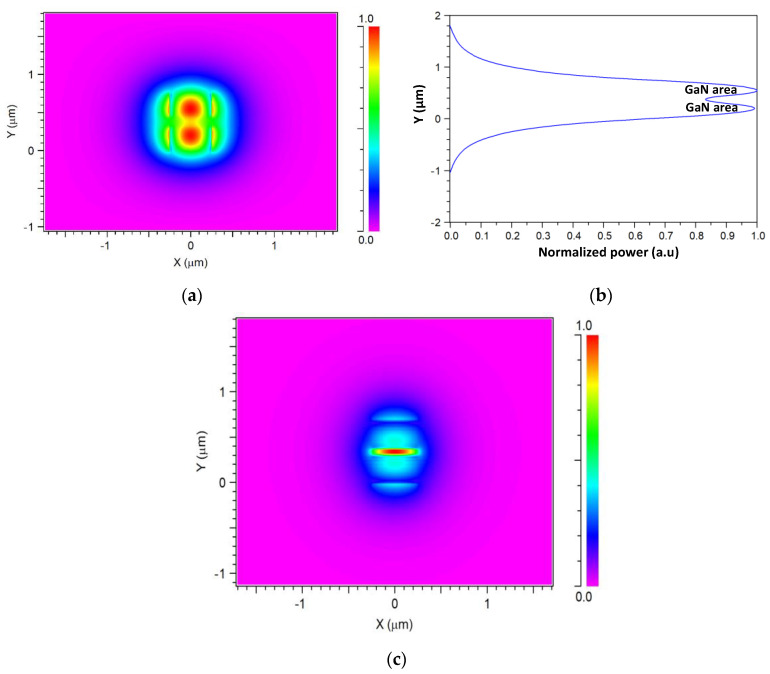
Fundamental intensity profile. (**a**) Transverse electric (TE) mode x–y cross section. (**b**). TE mode normalized intensity as a function of the y-axis. (**c**). TM mode x–y cross section.

**Figure 5 nanomaterials-10-02338-f005:**
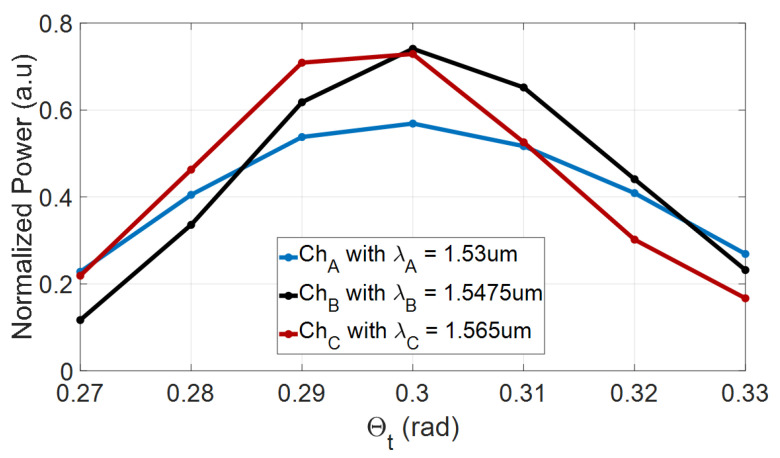
Normalized output power as a function of θ_t_ for the operating wavelengths.

**Figure 6 nanomaterials-10-02338-f006:**
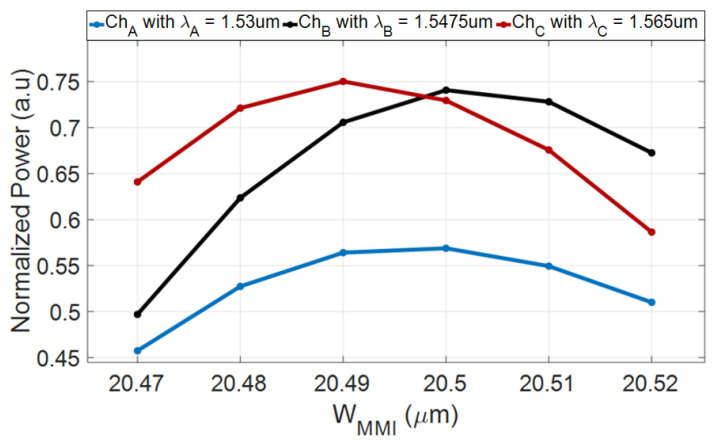
Normalized output power as a function of W_MMI_ (blue color for Ch_A_, black color for Ch_B_, and red color for Ch_C_).

**Figure 7 nanomaterials-10-02338-f007:**
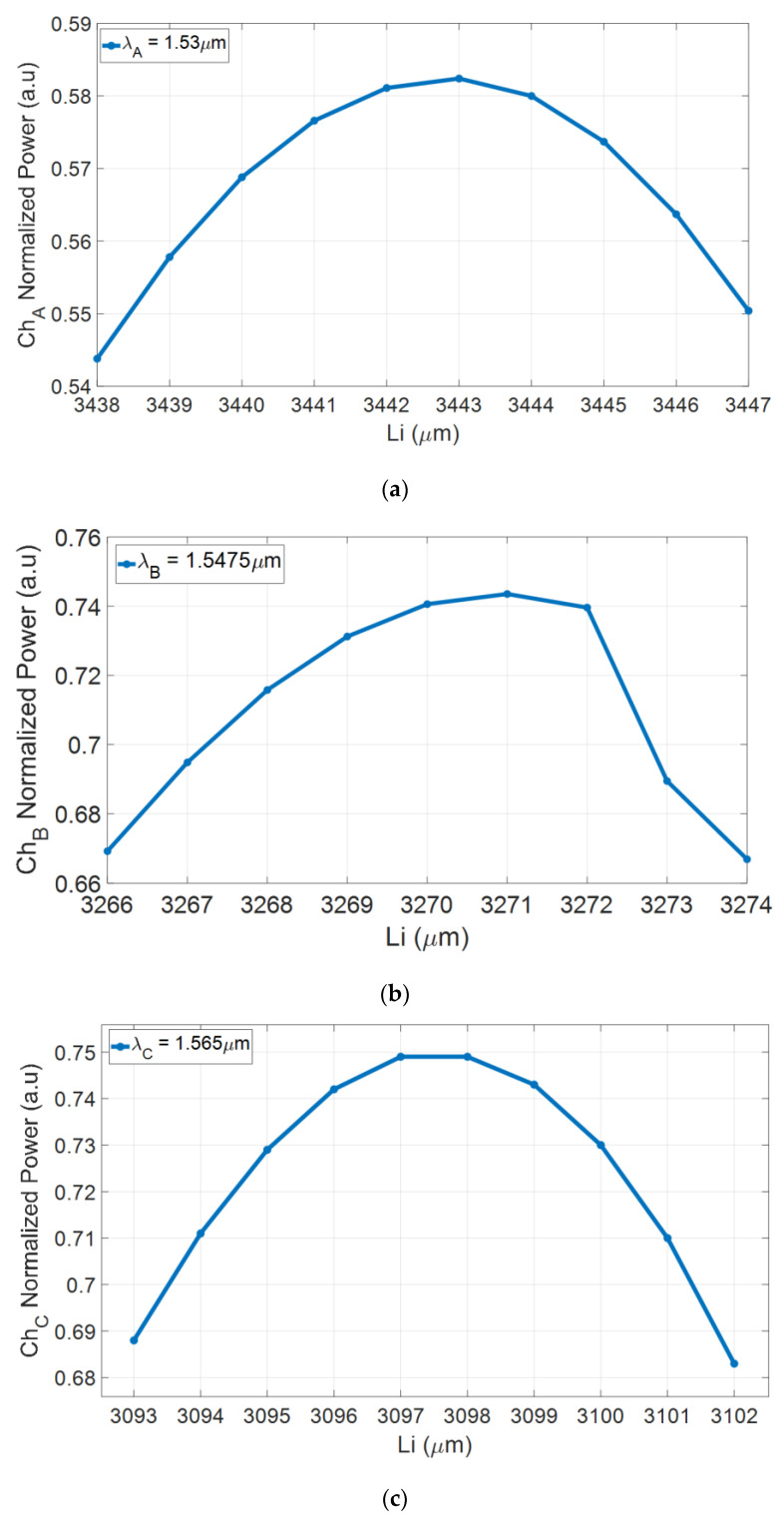
Normalized output taper power as a function of L_i_ (**a**), L_A_ (Ch_A_), (**b**) L_B_ (Ch_B_), and (**c**) L_c_ (Ch_C_).

**Figure 8 nanomaterials-10-02338-f008:**
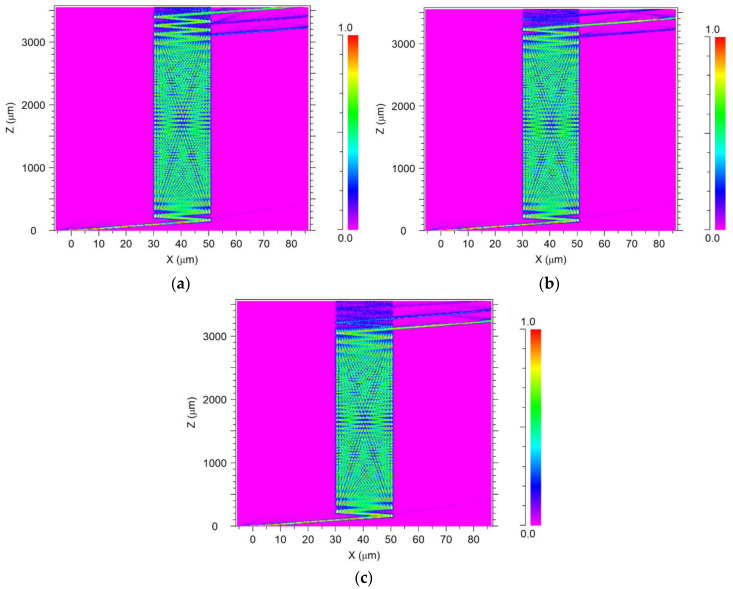
Light intensity profile of the AMMI 1 × 3 demultiplexer at the x–z plane. (**a**) 1.53 μm, (**b**) 1.5475 μm, and (**c**) 1.565 μm.

**Figure 9 nanomaterials-10-02338-f009:**
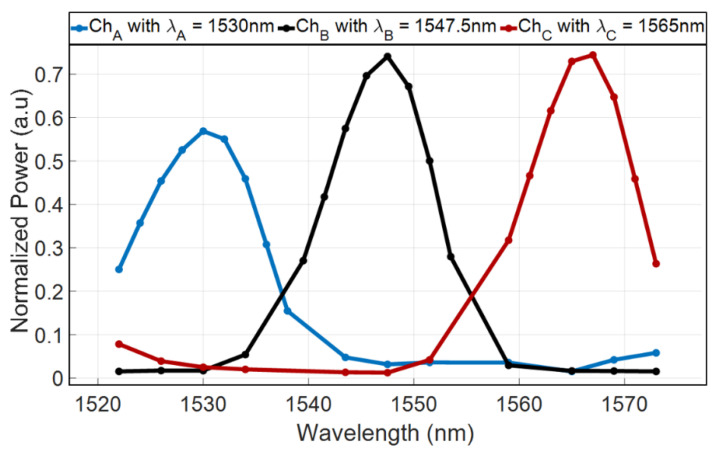
Optical spectrum around the C-band.

**Figure 10 nanomaterials-10-02338-f010:**
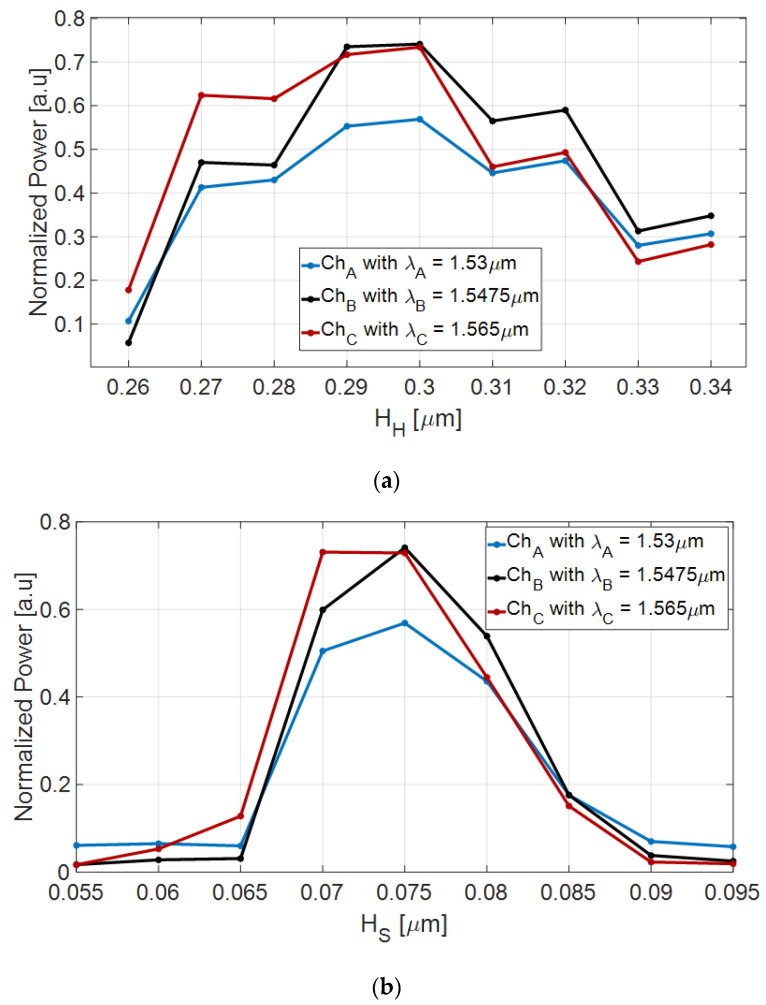
Normalized power at the output tapers as a function of the slot waveguide thickness. (**a**) H_H_ and (**b**) H_S_.

**Figure 11 nanomaterials-10-02338-f011:**
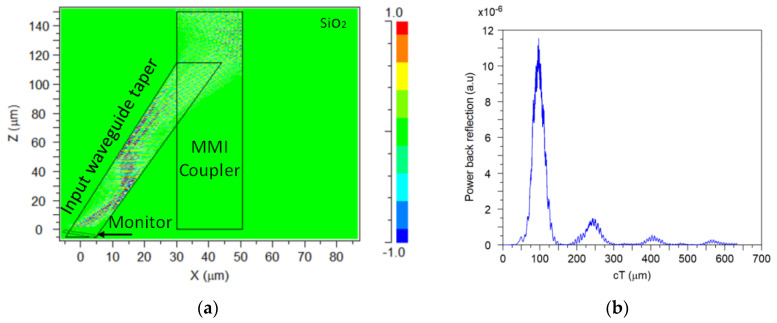
Power back reflection: (**a**) finite difference time domain (FDTD) simulation image and (**b**) power back reflection as a function of cT at 1.53 µm wavelength.

**Table 1 nanomaterials-10-02338-t001:** Optimized design length values for the three-channel angled multimode interference (AMMI).

Ch_i_	λ_i_ (μm)	L_i_ (mm)
Ch_A_	1.53	3.44
Ch_B_	1.5475	3.27
Ch_C_	1.565	3.095

**Table 2 nanomaterials-10-02338-t002:** Refractive indices and amplitudes ratio (AR) values for the operating wavelengths.

λ (μm)	n_H_	n_S_	AR
1.53	2.3173	1.44	2.589
1.5475	2.3169	1.44	2.588
1.565	2.3165	1.44	2.587

**Table 3 nanomaterials-10-02338-t003:** Values of the insertion loss (IL), crosstalk, and FWHM for the operational wavelengths.

λ (nm)	1530	1547.5	1565
Channel Notation	Ch_A_	Ch_B_	Ch_C_
Crosstalk (dB)	−12.79	−12.67	−15.62
FWHM (nm)	13.69	12.01	12.2
Insertion Losses (dB)	2.44	1.31	1.36

**Table 4 nanomaterials-10-02338-t004:** Power back reflection loss values for the operating wavelengths.

**Wavelength (µm)**	1.53	1.5475	1.565
**Power Back reflection loss (dB)**	48.76	47.85	47.64

**Table 5 nanomaterials-10-02338-t005:** Comparison between key characteristics of various AMMI demultiplexer types. SOI, silicon on insulator; WT, waveguide technology; NOC, number of channels; CS, channels spacing; IL, insertion losses; XT, best crosstalk; BR, back reflection.

AMMI Demultiplexer Type [5,31]	WT	NOC	CS [nm]	Footprint [mm^2^]	IL [dB]	XT [dB]	BR [dB]	Year
SOI	Rib	4	21	0.0145	~2	−25	Not stated	2011
N-rich Silicon nitride	Strip	4	20	0.034	~1.5	−20	Not stated	2018
This device	Slot-waveguide	3	17.5	0.0758	~2.4	−15	−48	In this paper

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
