# Peer review of "A Three Demultiplexer C-Band Using Angled Multimode Interference in GaN–SiO2 Slot Waveguide Structures"

_nanomaterials, 2020, doi:10.3390/nano10122338_

Round 1
Reviewer 1 Report
In response to the original manuscript submitted, I pointed out that the results lack novelty and won't contribute to the community.
The authors have made only very minor changes in the revised version of the manuscript, without any noticeable improvement.
Thus, my opinion on the manuscript is unchanged.
Author Response
We thank the reviewer for his comment. The novelty is a new study that shows how to design AMMI device using slot-waveguide technology. The aim of the work is to contribute to the progress of the field of photonic integrated circuits and especially to VLC-WDM systems. The ultimate goal of this field is the fabrication of devices and their use as part of different types of photonic systems. Nevertheless, I’d like to stress that the proposal presented in this article is highly numerical and theoretical in character, and the improvements put forward could be applied to a great variety of devices. I’m convinced that the dissemination of these results will be valuable for a number of researchers working in the field. All the calculations used in this article have been performed using the highly accurate FVBPM based on commercial Rsoft-cad software. It is widely recognized that the Rsoft-cad software provides an excellent match with the measurements performed on the fabricated devices and it is to be expected extremely high reliability of the results presented in this work when contrasted with measurements on actual devices. Unfortunately, I don’t have access to fabrication facilities because of the covid-19. I have the software that is normally used in the industry to produce the lithographic masks used in the device manufacturing at foundries. Nevertheless, we also in the revised version added new results that show the solution of the TM polarization mode. This increase the ability to use this technology for realization demultiplexer with a higher rank level.
Reviewer 2 Report
I thank the authors for their changes, especially the inclusion of additional simulation data, which I believe significantly strengthens the results. I also appreciate the additional background references and introductory material pertaining to both horizontally-slotted waveguides and MOCVD deposition of GaN. However, I still have concerns about the described fabrication approach and material configuration in combination with large size of the device. In particular, while high-quality crystalline GaN waveguides can exhibit low propagation losses (0.6-2.5 dB/cm) ((1)), the losses are highly dependent on the crystal quality and can easily reach into the range of 20 dB/cm. As ref. [25] points out, the bottom layer of the waveguide can be made crystalline, but the described fabrication approach would result in an amorphous or polycrystalline second layer. Such a film would have significant optical absorption (~13dB/cm of material absorption ((2)) in addition to further losses from increased etch difficulty ((1))). Given the 3mm length of the device, I would estimate at least an additional ~4dB of insertion loss from implementing the described approach, significantly altering a key result. It is possible that this loss could be avoided by using an alternate fabrication approach, or that better empirical data exists for the system in question, in which case I urge the authors to include that information in the introduction and references. If that is not the case, I would ask the authors to include a section discussing these additional sources of loss and providing estimated total loss values. ((1)) Chen, H. et al., Low loss GaN waveguides at the visible spectral wavelengths for integrated photonics applications, Optics Express, Vol. 25, Issue 25, pp. 31758-31773 (2017) doi: 10.1364/OE.25.031758 ((2)) Al-Zouhbi, A. and Al-Din, N. S. Structural and optoelectronic properties of amorphous GaN thin films. Optical Review, 15(5), 251–254. (2008) doi:10.1007/s10043-008-0039-3Author Response
Thank you for your suggestions which are very helpful for improving the quality of the paper. In the revised manuscript, we added a new section (IV GaN Fabrication losses) that describes and explains how it is possible to fabricate the proposed device. Also, the new section estimated the fabrication losses for light propagation of 3.56, 3.39, and 3.215 mm for the proposed device as the reviewer suggested.
Reviewer 3 Report
Authors present a simulation-based investigation of an angled MMI on horizontal slot waveguide platform. The concept is relevant and the structure very interesting. However, there is for me a fundamental misunderstanding that requires an explanation better than the one tentatively given at the end of the manuscript: why using quasi-TE polarized light in a horizontal slot waveguide. It means fabricating a very complex structure without benefiting of the slot waveguide effect.
A fast simulation of the modes in the structure shows a very high slot effect for the quasi TM fundamental mode (which makes sense) and not for the TE mode.
I suggest the authors to redo the work for TM polarization and resubmit a corrected version of the study, which is beside this very nice.
Moreover, in a waveguide, I think it is better to consider the Imbert Fedorov effect instead of the Goos Hanchen shift which is valid only for plane wave (so not a guided mode!).
Finally, I recommend also to not link the data points and to explain the abrupt variations observes in some curves and not in some others (Fig. 7b, for instance).
Author Response
Thank you for your suggestions which are very helpful for improving the quality of the paper. In the revised manuscript, we added the mode solution of the TM polarization as shown in figure 5(c). However, this solution is not suitable for the AMMI configuration and this is why we used the TE mode solution which also supports the single-mode transmission. We are planning to utilize this information in future work to design a TM/TE AMMI that can be used to obtained more channels.
Also, we replaced the Goos Hanchen shift with the lmbert Fedorov effect and explain the abrupt variations in figure 7 as the reviewer suggested.
Round 2
Reviewer 2 Report
I thank the authors for their changes, which have addressed my previous comments. I believe the paper is suitable for publication in its current form.Reviewer 3 Report
Authors have responded clearly to previous reviews and comments.
Despite a typo (lmbert instead of Imbert), I suggest to accept the manuscript for publication.